# The Effect of Proinflammatory Cytokines on the Proliferation, Migration and Secretory Activity of Mesenchymal Stem/Stromal Cells (WJ-MSCs) under 5% O_2_ and 21% O_2_ Culture Conditions

**DOI:** 10.3390/jcm10091813

**Published:** 2021-04-21

**Authors:** Aleksandra Wedzinska, Anna Figiel-Dabrowska, Hanna Kozlowska, Anna Sarnowska

**Affiliations:** 1Mossakowski Medical Research Centre, Translational Platform for Regenerative Medicine, Polish Academy of Sciences, 02-106 Warsaw, Poland; awedzinska@imdik.pan.pl (A.W.); adabrowska@imdik.pan.pl (A.F.-D.); 2Mossakowski Medical Research Centre, Laboratory of Advanced Microscopy Techniques, Polish Academy of Sciences, 02-106 Warsaw, Poland; hkozlowska@imdik.pan.pl; 3Mossakowski Medical Research Centre, Stem Cell Bioengineering Unit, Polish Academy of Sciences, 02-106 Warsaw, Poland

**Keywords:** mesenchymal stem cells, WJ-MSCs, inflammation, acute infection, paracrine activity

## Abstract

Treatment with Mesenchymal Stem/Stromal Cells (MSCs) in clinical trials is becoming one of the most-popular and fast-developing branches of modern regenerative medicine, as it is still in an experimental phase. The cross-section of diseases to which these cells are applied is very wide, ranging from degenerative diseases, through autoimmune processes and to acute inflammatory diseases, e.g., viral infections. Indeed, now that first clinical trials applying MSCs against COVID-19 have started, important questions concern not only the therapeutic properties of MSCs, but also the changes that might occur in the cell features as a response to the “cytokine storm” present in the acute phase of an infection and capable of posing a risk to a patient. The aim of our study was thus to assess changes potentially occurring in the biology of MSCs in the active inflammatory environment, e.g., in regards to the cell cycle, cell migration and secretory capacity. The study using MSCs derived from Wharton’s jelly (WJ-MSCs) was conducted under two aerobic conditions: 21% O_2_ vs. 5% O_2_, since oxygen concentration is one of the key factors in inflammation. Under both oxygen conditions cells were exposed to proinflammatory cytokines involved significantly in acute inflammation, i.e., IFNγ, TNFα and IL-1β at different concentrations. Regardless of the aerobic conditions, WJ-MSCs in the inflammatory environment do not lose features typical for mesenchymal cells, and their proliferation dynamic remains unchanged. Sudden fluctuations in proliferation, the early indicator of potential genetic disturbance, were not observed, while the cells’ migration activity increased. The presence of pro-inflammatory factors was also found to increase the secretion of such anti-inflammatory cytokines as IL-4 and IL-10. It is concluded that the inflammatory milieu in vitro does not cause phenotype changes or give rise to proliferation disruption of WJ-MSCs, and nor does it inhibit the secretory properties providing for their use against acute inflammation.

## 1. Introduction

Mesenchymal stem/stromal cells (MSCs) have several unique properties that favour their use in therapy. One of the most important features is immunomodulative activity [1,2]. In the case of activation of the inflammatory process, MSCs indirectly interact with all types of cells of the immune system through the secretion of therapeutic factors that show antiapoptotic, angiogenic and immunosuppressive activity [3,4]. Moreover, the MSCs may also interact with the immune system via direct cell-cell contact with every type of immune system cell. The presence of MSCs in the environment helps regulate the survival of individual subpopulations of T helper (Th) lymphocytes. Additionally, MSCs interact with natural killer cells (NK cells) according to the state of activation of the latter, and/or with cytokines present in the environment [5]. In addition, the presence of mesenchymal stem cells prevents the induction of NK-cell cytotoxic activity. This relates directly to MSC-secreted factors such as indoleamine 2,3 dioxygenase (IDO) and prostaglandin E2 (PEG2) [6]. The same factors are involved in polarization of macrophages from the pro-inflammatory to the anti-inflammatory phenotype [7,8]. The secretory profile of MSCs is thus influenced by the microenvironment, which they reside on. Depending on the factors present in the microenvironment, mesenchymal cells may adopt a pro- or anti-inflammatory phenotype determined similarly to macrophages, as MSC1 and MSC2 [9]. These properties are even present in the cells in the resting form, with polarization depending on the stimuli to which they are exposed [10,11]. Unfortunately, there is not enough information that allow for predicting behavior of these cells in a pathological environment. Both the direct and indirect influence of MSCs on the components of the immune system legitimize their application as an immunomodulating factor in clinical trials.

The use of MSCs in treating autoimmune diseases such as lupus erythematosus and rheumatoid arthritis has resulted in a decrease of inflammatory markers and improvement of symptoms [12]. Numerous studies have also demonstrated that by reducing persistent inflammation that coexist and exacerbate neurodegenerative processes, MSCs treatment might be effective in Alzheimer disease [13] and Parkinson’s disease [14,15] and amyotrophic lateral sclerosis (ALS) [16].

In addition to a widely described anti-inflammatory effect in chronic diseases, the beneficial impact of MSC has also been shown in acute inflammatory conditions accompanying injuries or infections. Intravenous administration of MSCs from adipose derived tissue in the subacute phase of intracerebral hemorrhage improves patients’ neurological condition by suppression of acute neuroinflammation [17]. MSCs have also become a promising tool for the treatment of virus-associated diseases, such as immunologic abnormality followed by infection with Human Immunodeficiency Virus (HIV) or acute lung injury (ALI) caused by influenza virus. On the other hand, MSCs have been shown to be susceptible to avian influenza virus infections, which led to losing of their immunoregulatory activity and apoptosis [18]. Therefore the interplay between MSC and proinflammatory environment can be a double-edge sword, which may lead to therapeutic benefits but also to cell death or mutagenesis.

While the mechanism of MSCs therapeutic action following transplantation is already known, more studies are needed to address the impact of a pathological environment on MSCs, morphology, proliferation and migration. From the clinical point of view, a key question is whether the cells in contact with strong stress factors will proceed to apoptosis or to excessive uncontrolled proliferation resulting in malignancy.

The maladaptive cytokine release in response to infection or other stimuli called “cytokine storm” leads to a breakdown of the normal immune response and to pathological changes in stem cell microenvironment. It includes the increase of reactive oxygen species (ROS) production/oxidative stress and/or early cells’ senescence [19]. One of the drawbacks of the latter is a loss of tissue repair capacity due to diminishing self-renewal (pool preservation impact) and differentiation (tissue imbalance) caused by the cell cycle arrest. The other is a microenvironment modulation by senescent MSC due to secretion of pro-inflammatory and matrix-degrading molecules, which, if escalated, might have a significant local or systemic impact on overall organism homeostasis.

In the presence of many immunoactive molecules, the natural immunological border may collapse. The The Tumor Necrosis Factor α (TNFα), Interefon γ (IFNγ), Interleukin 1 (IL-1), Interleukin 2 (IL-2) and Interleukin 6 (IL-6) areamong the cytokines significantly elevated in inflammatory diseases with pathophysiological similarities, such as COVID-19, Cytokine release syndrome, Secondary hemophagocytic lymphohistiocytosis and Immune Reconstitution Inflammatory Syndrome [20]. At the same time, three of them—TNFα, IFNγ or IL-1—are required for MSCs activation needed to modulate immune response [21].

TNFα is one of the main factors involved in tumor progression and has a crucial role in epithelial mesenchymal transition (EMT). TNFα and IFNγ—as the two top inflammatory mediators acting in synergy and used in in-vitro models, represent a “lethal” combination that massively up-regulates inflammatory responses. Together they induce the production of superoxide anions, as well as oxygen and nitrogen radicals. TNFα is present at the highest concentration in the core of the infection, whereas the IFNγ gradient linked to NK-cell infiltration is highest at the periphery [22].

IL-1 is a pleiotropic cytokine responsible for inducing fever through activation of the hypothalamus-pituitary-adrenal (HPA) axis in virial infections [23]. IL-1β activates mast cells and induces histamine production, increasing membrane permeability [24].

Based on previous experiments, it seems that concentration and the time of exposure to the above-mentioned cytokines may influence the MSCs’ therapeutic effects in two contrasting ways—by “licencing”/activating the MSCs, or inducing MSC death through apoptosis, necroptosis, or autosis [25].

To analyze the effect of the inflammatory environment on the human MSC, we selected mesenchymal stem/stromal cells derived from Wharton’s jelly/umbilical cord stroma (WJ-MSC). Due to a high availability of the source tissue, lack of invasiveness in isolation and low immunogenicity [26,27], WJ-MSC are most often used in current clinical trials, in an allogenic-system context.

Since the oxygen concentration is a key factor responsible for maintaining a proper homeostasis and influence the inflammation [28,29,30,31], WJ-MSC in our model were cultivated under 21% or 5% O_2_ oxygen conditions.

## 2. Material and Methods

### 2.1. The Isolation and Culture of WJ-MSCs

All study protocols were approved by the Ethics Committee of Warsaw Medical University. Samples of human umbilical cord were collected and processed for MSC isolation using mechanical fragmentation techniques. The cords (20 cm lengths of the fetal part) were rinsed with sterile phosphate-buffered saline (PBS; Gibco, Thermo Fisher Scientific, Waltham, MA, USA)) and supplied with a cocktail of antibiotics comprising penicillin, streptomycin and amphotericin B (1:100, Gibco, Thermo Fisher Scientific, Waltham, MA, USA)). They were then cut with a sharp, sterile blade into 2 mm slices. Using a biopsy punch (Miltex, GmbH, Viernheim, Germany), small 2 mm^3^ fragments of Wharton’s jelly matrix were then removed from umbilical cords and transferred to culture dishes. They were maintained subsequently in Dulbecco’s Modified Eagle’s Medium (DMEM, Thermo Fisher Scientific, Waltham, MA, USA) with added 10% human platelet lysate (Macopharma, Tourcoing, France) and the aforesaid cocktail of antibiotics at 100 mg/mL). The temperature maintained was 37 °C, with the 5 or 21% O_2_ levels referred to, and 5% CO_2_ in a humidified atmosphere. In all experiments WJ-MSCs used were from 3 patients, with cells used for passage 3 to 5 (P3–P5).

### 2.2. Proinflammatory Cytokines

To evaluate activation of WJ-MSCs in vitro, cells were cultured using a standard procedure reaching 70% confluence. Medium was then replaced with another lacking human platelet lysate in order for external growth factors and chemokines to be eliminated. Culture medium then received proinflammatory cytokines: IFNγ and TNFα (1:1—5 ng/mL, 12.5 ng/mL or 25 ng/mL) or IL-1β (10 ng/mL, 25 ng/mL or 50 ng/mL), with cell morphology, proliferation, cell cycling and migration evaluated 24 h on from the start of the incubation. To analyze paracrine activity of WJ-MSCs, the supernatant was harvested and assessed by way of Luminex assay. All experiments were prepared following in scheme (Figure 1).

### 2.3. FACS Analysis of Surface Markers

Analysis of surface markers characteristic for MSCs was achieved using the Human MSC Analysis Kit (Becton Dickinson, Franklin Lakes, NJ, USA). Following the manufacturer’s protocol, each tube received fluorochrome-conjugated antibodies directed against APC CD73, FITC CD90, and PerCPCy_5.5 CD105 (positive markers) and PE CD34, PE CD11b, PE CD19, PE CD45, and PE HLA-DR (negative markers), prior to incubation for 30 min in the dark at room temperature. Cell analysis was performed using FACSDiva software with the FACSCanto II program (Becton Dickinson, New Franklin Lakes, NJ, USA).

### 2.4. Mesodermal Lineage Differentiation

#### 2.4.1. Osteogenesis

When cells reached the proper confluence, the medium was replaced to differentiation medium from an Osteogenesis Differentiation Kit (Gibco, Thermo Fisher Scientific, Waltham, MA, USA). After 21 days of differentiation, cells were fixed with 4% PFA for 30 min and washed with PBS. Fixed cells were washed twice with distilled water and then stained with 2% Alizarin Red S (Sigma-Aldrich, Saint Louis, MO, USA To stain the cells, dye solution was applied for 3 min prior to rinsing with distilled water.

#### 2.4.2. Chondrogenesis

Following standard culture, WJ-MSCs were detached by Accutase and centrifuged, prior to the removal of the culture medium. Cells were then seeded as a 5 μL drop/well in a 24-well plate and incubated for 60 min at 37 °C. A differentiation medium was added and cells were cultured for 14 days. They were then fixed with 4% PFA, with chondrogenesis confirmed through Alcian blue staining for the presence of cartilage glycosaminoglycans. For this purpose, previously prepared cells were stained with a 1% solution of Alcian blue (Sigma-Aldrich, Saint Louis, MO, USA) in 0.1 N HCl, prior to incubation for 30 min at RT. Excess dye was then rinsed off with 0.1 N HCl, prior to neutralization with distilled water.

#### 2.4.3. Adipogenesis

When cells reached the proper confluence, the medium was replaced with differentiation medium from the Adipogenesis Differentiation Kit (Gibco, Thermo Fisher Scientific, Waltham, MA, USA).). After 14 days of differentiation, cells were fixed with 4% PFA for 30 min and washed with PBS, and then 60% isopropanol was added for 5 min.

Staining was achieved with 99% isopropanol plus Oil Red O (Sigma-Aldrich, Saint Louis, MO, USA). The resulting solution was diluted in distilled water (3:2). Isopropanol was removed and the stain added after 10 min of incubation of the cells for 5 min to verify the positive effect of differentiation.

### 2.5. Immunocytochemistry

For immunofluorescence staining, the cell culture was fixed in 4% PFA (Sigma-Aldrich, Saint Louis, MO, USA) and 0.2% Triton X-100 (Sigma-Aldrich) in PBS for 15 min at RT. Nonspecific reactions were blocked with 10% goat serum (GS, Sigma-Aldrich) in PBS for 60 min at RT. To examine the present of mesenchymal markers primary antibodies against fibronectin: polyclonal anti-fibronectin antibody (rabbit), IgG (H + L), 1:500 (Sigma-Aldrich) and vimentin: monoclonal anti-vimentin antibody, IgG1, 1:200, (Dako) was added (Appendix A). After overnight incubation, a secondary antibody (Alexa Fluor 488, 1:1000; Invitrogen, Thermo Fisher Scientific, Waltham, MA, USA) was added for 1 h at RT (Appendix A). Cell nuclei were counterstained with Hoechst 33258 (1:150). Finally, labelled cells were analyzed under a confocal laser scanning (LSM 780) microscope.

### 2.6. LDH Leakage Assay

Lactate dehydrogenase (LDH) release is an indicator of outer cell-membrane injury. The amount of LDH release in the medium after cell incubation with proinflammatory cytokines was analyzed. WJ-MSCs were seeded on to 96-well cell-culture plates at an initial density of 3 × 10^3^ cells/cm^2^; and cultured up to 70% of confluence. The medium was then replaced to a medium containing IFNγ and TNFα (1:1—5 ng/mL, 12.5 ng/mL or 25 ng/mL) or IL-1β (10 ng/mL, 25 ng/mL or 50 ng/mL). After 24 h of incubation, 50 μL of medium were transferred to a new 96-well plate and an LDH working solution was added to each well, in line with the producer’s instructions (LDH Cytotoxicity Assay Kit, ThermoFisher Scientific, Waltham, MA, USA). Plates were incubated for 20 min at room temperature (RT) in the darkness. The reaction was completed by adding 50 μL of stop solution. Absorbance was measured using a wavelength of 490 nm in an enzyme-linked immunosorbent assay leader (Spark 10M, Tecan, Mannendorf, Switzerland).

### 2.7. Cell Viability and Proliferation

Cell viability was evaluated using a MTT (3-(4,5-dimethylthiazol-2-yl)-2,5-diphe-nyltetrazolium bromide) assay. The cells were seeded on 96-well cell-culture plates at an initial density of 3 × 10^3^ cells/cm^2^. After one day, the medium was replaced to another containing cytokines, and cells were incubated for the next 24 h. 10 μL of MTT salt was added to the wells and incubated for 3 h. 25 μL of solution was left in the wells, and 50 μL of DMSO added. Absorbance was measured in an immunosorbent assay leader (Spark 10M, Tecan, Mannendorf, Switzerland).

Differences in cell proliferation were determined by calculating population doubling time (PDT). WJ-MSCs with all proinflammatory factors were cultured in 5% or 21% O_2_. Then, reaching 80% confluence, cells were collected and counted using a Bürker chamber. PDT was calculated based on the total cell number at each passage using the formula (t − t0) × log 2/(log N − log N0), PDs = log(N0/Ni)/log 2; CPDs = PDs1 + PDs2 + PDs3 + … + PDsn; where t − t0 is the culture time (days), N is the number of harvested cells, and N0 is the initial number of cells.

### 2.8. The Cell Cycle

The cell cycle was evaluated using flow cytometry. WJ-MSCs at a 70–80% confluence were incubated for 24 h, with INFγ + TNFα at a concentration of 5 ng/mL, or else INFγ and TNFα at a concentration of 12.5 ng/mL, or IL-1β at a concentration of 50 ng/mL. Next WJ-MSCs were detached by accutase, washed with PBS and fixed with 70% cold ethanol. The cells were then centrifuged, rinsed with PBS, and resuspended in a staining solution containing 100 µg/mL of RNase A (PureLink RNase A; Invitrogen, Thermo Fisher Scientific, Waltham, MA, USA), 0.1% of Triton X-100 (Sigma Aldrich, Saint Louis, MO, USA) and 10 µg/mL of propidium iodide (PI, Invitrogen). Cells were incubated in the dark for 30 min at RT. DNA content was determined by flow cytometry (BD FACSCanto II) and the percentage of cells in different phases of the cell cycle was assessed. A minimum of 10^4^ events per sample were acquired.

### 2.9. Cell Migration

The influence of inflammatory factors on WJ-MSC migratory activity was assessed by scratch assay. WJ-MSCs were cultured in 24-well plates. After 80% confluency, a scratch was made with a 200 μL pipette tip to simulate a wound. The well was washed with PBS buffer prior to culture medium with selected groups of proinflammatory stimulants being added. After 24 h incubation, cells were fixed in 4% paraformaldehyde, and nuclei stained with Hoechst 33342. Quantitative analysis of cells in the scratch area was achieved.

### 2.10. Chemotaxis

The impact of inflammatory factors on WJ-MSC migration was analyzed with μSlide plates for chemotaxis evaluation (Ibidi, Gräfelfing, Germany).

For this purpose, 6 μL of WJ-MSC suspension (cell density—1.6 × 10^6^ cells/mL) was injected into the central chamber and incubated for 3 h at 37 °C to allow the cells to adhere to the plate. Two types of medium were then added to the lateral chambers; i.e., control medium without platelet lysate (left side) and selected stimulants (right side) (Figure 2). As a positive control, DMEM with bFGF at 20 ng/mL (cell attractant) was used. Over a 24-h, cell activity was observed under an AxioObserver Z.1 microscope (Carl Zeiss, Oberkochen, Germany) in transmitted light. Cell locations were recorded every 10 min. During the experiment, 37 °C, 5% CO_2_ and 21% O_2_ conditions were maintained. The obtained image was analyzed qualitatively using the Zen 2.0 program (Zeiss, Oberkochen, Germany)), while quantitative analysis used ImageJ and Chemotaxis Migration Tools (Ibidi, Gräfelfing, Germany). The results are present as *p*-value Rayleigh test).

### 2.11. The WJ-MSC Secretome Following Treatment by Simulating Proinflammatory Cytokines Stimulation

Post-culture-medium analysis was performed using the MagneticLuminex^®^ Assay: Human Premixed Multi-Analyte Kit (R&D Systems, Minneapolis, MN, USA).

Cells in the same density were cultured up to 70–80% confluency prior to exchange of the medium for the one containing proinflammatory cytokines. After 24 h, the culture medium was collected and centrifuged for 5 min. at 800× *g*. The obtained media were concentrated using Spin-X UF 6 concentrator columns (Sigma-Aldrich) with protein cleavage above 5 kDa and centrifuged for 15 min, at 1800× *g*. Total protein was measured using the Bradford method to verify that the protein concentration range fell within the used kit’s standard curve. The samples used for analysis were given in the same volume. Factor analysis was performed using the MagneticLuminex^®^ Assay: Human Premixed Multi-Analyte Kit ((R&D Systems, Minneapolis, MN, USA) in line with the manufacturer’s protocol. The concentration of IL-4, IL-10, CCL2, CXCL10, IL-6 and IL-12 in the collected medium was determined, and estimated with a Luminex Bio-Plex^®^ 200 System (Bio-Rad, Harcules, CA, USA) device.

### 2.12. Organotypic Hippocampal Culture (OHC)

Organotypic slice culture was obtained according to the Stoppini method, modified in our lab. Male and female 1-week-old Wistar rat pups were provided by Mossakowski Medical Research Centre Animal Breeding House. At the time of sacrifice, pups were cooled on ice and decapitated. Brains were extracted, and rat hippocampi were isolated, sectioned into 400-μm slices using a McIlwain tissue chopper (Ted Pella, Poznan, Poland), and transferred to Millicell CM (Millipore, Warsaw, Poland) membranes placed in six-well plates (Nunc; Thermo Fisher Scientific, Waltham, MA, USA). The culture medium was composed of neurobasal medium (75%; Gibco), Nutritional N2 (1:10; Gibco), B27 (1:100; Gibco), and HBSS, HEPES, glucose, l-glutamine, and antibacterial– antimycotic solution (as mentioned before). After two weeks the slices were used for the Oxygen-Glucose Deprivation (OGD) procedure and co-culture experiments.

### 2.13. Oxygen-Glucose Deprivation in OHC Slices in a Transwell Co-Culture Model System

For the OGD procedure, the membranes with hippocampal slices were transferred to an anaerobic chamber immersed inserted in Krebs–Ringer solution (Sigma-Aldrich) supplemented with 10 mM mannitol (as the glucose substitution; Sigma-Aldrich) and saturated with 95% N_2_ and 5% CO_2_. The cultures were kept in an oxygen-free atmosphere (95% N_2_/5% CO_2_) for 40 min in order to mimic an ischemic injury. Simultaneously, the control slices were maintained in glucose-containing Krebs-Ringer solution in normoxic conditions. To evaluate MSC-immunomodulatory effect, hippocampal slices were transferred immediately after injury under the cell culture and co-cultured for 24 h with WJ-MSCs (at 80% confluency) or WJ-MSC pre-treated with cytokines in serum-free medium. The slices were then fixed with 4% paraformaldehyde (PFA; Carl Roth, Warsaw, Poland) for 30 min. at room temperature (RT) and rinsed with PBS. Slices were incubated in the blocking buffer for 1 h at RT. Fixed slices were labelled overnight at 4 °C with primary antibody against rat macrophages/microglia marker ED-1. Hippocampal slices were rinsed three times with PBS and incubated for 1 h at RT with Alexa 488 fluorochrome (Invitrogen, Thermo Fisher Scientific, Warsaw, Poland) in PBS. Stained slices were viewed by the confocal laser scanning microscope (LSM 780; Carl Zeiss, Oberkochen, Germany) and analyzed by ZEN 2012 software (Carl Zeiss, Oberkochen, Germany).

### 2.14. Statistical Analysis

Data were analyzed using one-way analysis of variance (ANOVA) with the Tukey’s post-hoc test from GraphPadPrism 7.0. Cytokine analysis was performed using two-way analysis of variance (ANOVA). Differences associated with a *p*-value ≤ 0.05 were defined as significant statistically.

## 3. Results

### 3.1. The Mesenchymal Character of Cells Isolated from Wharton’s Jelly

The mechanical method of WJ-MSC isolation allowed for obtaining a large number of cells with high migration potential and a morphology characteristic for young cells e.g., elongated, cylindrical shape, small and round, with a single, centrally-located nucleus (Figure 3A). Following adhesion on a plastic dish, long-term cultivation led to the changes in cells morphology. WJ-MSC became more flattened, larger and amoeba-shaped, indicating the senescence of cell population (Figure 3B). The WJ-MSCs obtained for further analysis complied with the recommendation of the ISCT as: adherence to plastic, specific surface antigens and multipotent differentiation potential [29].

Cytometric analysis of the surface markers characteristic for mesenchymal stromal/stem cells (CD73+, CD90+ and CD105+) confirmed the expression of all the above mentioned markers in approx. 99%, while approx. 1% of the population was positive for hematopoietic markers (Figure 3C). WJ-MSCs possessed the ability to differentiate into osteocytes and chondrocytes, as was confirmed in staining with Alizarin S and Alcian blue respectively (Figure 3E,F). After differentiation toward adipocytes, regular red fat drops were present in the cytoplasm of cells in Oil Red staining (Figure 3D). Differentiation towards adipocytes was not as effective as in the other directions. Immunocytochemical analysis of the cells confirmed the presence of cytoskeleton proteins: vimentin (Figure 3G) and fibronectin (Figure 3H).

### 3.2. Influence of Pro-Inflammatory Cytokines on WJ-MSC

In order to find the concentration of pro-inflammatory factors that will not evoke immediate cell death or significantly reduced viability, the assessment of the outer cell membrane integrity and cell viability was performed for selected factors and concentrations.

Lactate dehydrogenase (LDH), as a soluble cytosolic enzyme, is released into culture medium upon cell death, due to damage to the plasma membrane. The effect of pro-inflammatory cytokines on WJ-MSCs was estimated by reference to increased LDH activity in culture supernatant, which was proportional to the number of lysed cells. There were no significant differences in the amounts of lactate dehydrogenase released from cells exposed to cytokines under both 21% and 5% oxygen conditions (Figure 4A). While the co-addition of IFNγ and TNFα was associated (in a dose-dependent manner) with a slight increase in the lactate dehydrogenase content in the medium, in the concentration of 25 ng/mL, these results did not achieve statistical significance. Such results point the preserved integrity of the outer cell membrane—with a lack of cytotoxicity of selected stimulants on WJ-MSCs, regardless of their concentration, or the aerobic conditions present in culture.

The next step was to assess the rate of cell viability under such conditions at both concentrations of oxygen (atmospheric 21% vs. 5%). The MTT assay is used to measure cellular metabolic activity as an indicator of cell viability. Cells active metabolically contain NAD(P)H-dependent oxidoreductase enzymes that reduce tetrazolium salt (3-(4,5-dimethylthiazol-2-yl)-2,5-diphenyltetrazolium bromide or MTT) to formazan. The assay revealed no statistically significant differences in cell viability in the two tested oxygen concentrations (Figure 4B). In all tested concentrations TNFα and IL-1β did not result in a significant reduction of cell viability, although IFNγ in the 5% oxygen conditions and at a concentration of 10 ng/mL decreased the cells viability up to 71% ± 4.

According to the literature reports describing a synergistic effect of IFNγ and TNFα, the viability after combined proinflammatory induction was assessed.

Once levels of cytokines not inducing the immediate cell death through loss of membrane integrity or cell apoptosis had been determined, cell fate analysis under sub-lethal stress conditions were commenced with IFNγ and TNFα together, at concentrations of 12.5 and 25 ng/mL respectively, as well as (separately) of IL-1β at a concentration of 50 ng/mL.

The cells treatment with proinflammatory cytokines in fixed concentrations neither induced changes in proportion of typical mesenchymal surface markers nor in their morphology.

The proliferation potential was analyzed using population doubling time (PDT) setting. The rate of cell division was assessed under 21% and 5% oxygen conditions. The cells cultured in 5% oxygen concentration indicated 2-fold higher proliferative potential than cells cultured in 21% oxygen (1.29 ± 0.07 days vs. 2.3 ± 0.32 days) (Figure 4C). However proliferation rate was not changed significantly in the presence of the pro-inflammatory cytokines.

IFNγ and TNFα in concentration 5 ng/mL, 12.5 ng/mL and IL-1β in concentration 50 ng/mL were selected for subsequent determinations. The selected concentrations did not affect cell viability (MTT) but caused a slight increase in the amount of LDH in the medium (minimal damage to the cell membrane–LDH leakage).

### 3.3. The Influence of Inflammatory Factors on the Cell Cycle

The effect of cytokines on the cell proliferation was determined by estimating the percentage of the cells in each phase of the cell cycle (Figure 5A,B). Cell cycle analysis revealed no differences in the percentages of the cells in each phase. The proinflammatory cytokines did not act to inhibit the life functions of cells. Under 5% oxygen conditions, an increased number of cells in the S phase was observed in all tested variants, indicating an increased capacity for cell division as compared with cells maintained under atmospheric oxygen conditions (21%) (Figure 5C).

### 3.4. Analysis of WJ-MSC Migration and Chemotaxis

Under both oxygen conditions, the migration potential of WJ-MSCs was evaluated using the scratch test following stimulation with IFNγ + TNFα (both cytokines at 5 ng/mL), IFNγ + TNFα (at 12.5 ng/mL) and IL-1β (at 50 ng/mL). The results were presented as a percentage of the control under applied aerobic conditions (21% or 5%).

Lower oxygen concentration (5%) increased cell migratory activity. Moreover, the presence of cytokines in the environment, especially in higher concentrations, enhanced this effect (Figure 6A). Only IFNγ + TNFα at the low concentration (5 ng/mL) under 5% oxygen condition resulted in a decreased cell migration by 36.67% compared to the control maintained under the same oxygen concentration. The higher concentration of IFNγ + TNFα (12.5 ng/mL) exert a non-significant increase in a migratory potential (104.6% in 21% O_2_ compared with 115.9% with 5% O_2_). Under both oxygen conditions, it was IL-1β that activated the most cell migration.

To analyze the chemotaxis of WJ-MSCs toward the inflammatory core, measurement in real-time in a 2D environment was performed using a µSlide plate (Figure 6B). WJ-MSCs did not exhibit motility in the control environment (negative control without stimulants in the medium). However, slightly increased migration (non-statistically important) towards the bFGF attractant was observed (positive control). Significant focused migration was observed where inflammatory factors were present in the environment. The strongest effect ensued when IL-1β was added to the medium, with 80% of analyzed cells showing cytokine taxis and active migration. The cells demonstrated the longest forward migration, and high-speed motility. Addition of IFNγ + TNFα (5 ng/mL) evoked also a significant increase in the migration potential, albeit to a more limited extent than with IL-1β.

### 3.5. The Secretory Profile of WJ-MSCs

The secretory properties of WJ-MSCs in a pro-inflammatory environment were analyzed using Luminex technology, in relation to the cyto- and chemokines involved most often in the acute inflammation, i.e., IL-4, IL-6, IL-10, IL-12, CCL2 and CXCL10 (Figure 7).

When cultured in control conditions (irrespective of the oxygen conditions), WJ-MSCs were not found to secrete IL-6, IL-10, IL-12 or CXCL10. However, in the baseline state small amounts of IL-4 and CCL2 were released, more significant in 21% O_2_.

In the presence of pro-inflammatory factors in the environment, cytokine and chemokine secretion increased comparably under both aerobic conditions. The increase in secretion was not dependent on the concentration of the inflammatory factor.

Stimulation with IFNγ + TNFα at both tested concentrations was followed by significant secretion (in both oxygenation conditions) of IL-10 (for 21% O_2_ respectively 75 ± 7 and 76 ± 5 pg/mL and for 5% O_2_: 75 ± 6 and 74 ± 6 pg/mL) and IL-4 (43 ± 6; 48 ± 3; 42 ± 3 and 46 ± 4 pg/mL); and for CCL2 chemokines (21% O_2_: 23,425 ± 6291 and 24,312 ± 4420 pg/mL; and for 5% O_2_: 25,929 ± 1551 and 20,985 ± 3115 pg/mL) by WJ-MSC. Differences between these results were significant.

With stimulation of WJ-MSCs by IL-1β at a concentration of 50 ng/mL, we also observed greater releases of anti-inflammatory chemokines and cytokines into the medium than in the control not exposed to the inflammatory factors.

Equally, IL-1β was the only tested factor to evoke CXCL10 secretion in MSCs (21% O_2_: 588 ± 407 pg/mL; 5% O_2_: 546 ± 401 pg/mL). IL-1β activated WJ-MSCs to secrete IL-6 to a significantly greater extent than the other two stimulants (21% O_2_: 9068 ± 3068 pg/mL and 5% O_2_: 11717 ± 334.5 pg/mL vs. IFNγ + TNFα at 5 ng/mL, with 21% O_2_: 1851 ± 1292 pg/mL and 5% O_2_: 2678 ± 1009 pg/mL; and IFNγ + TNFα at 12.5 ng/mL with 21% O_2_: 4128.6 ± 2802 pg/mL; and with 5% O_2_: 2955 ± 1411 pg/mL). At the same time, the activation of cells as regards IL-12 secretion was more limited than with IFNγ and TNFα stimulation at both concentrations (with 21% O_2_: 289 ± 162 pg/mL and with 5% O_2_: 365 ± 89 pg/mL vs. IFNγ + TNFα at 5 ng/mL, with 21% O_2_: 519 ± 59 pg/mL; with 5% O_2_: 562 ± 52 pg/mL; and IFNγ + TNFα at 12.5 ng/mL—with 21% O_2_: 585 ± 41 pg/mL; and with 5% O_2_: 562 ± 44 pg/mL).

### 3.6. Immunomodulatory Potential of WJ-MSC under the Co-Culture with Hippocampal Slice Culture

The current results demonstrate that exposure of organotypic hippocampal slice cultures to oxygen-glucose deprivation leads to significant immuno-response of injured brain tissue and robust microglia activation (Figure 8). The indirect (separated with membrane) co-culture with WJ-MSC or pre-treated with cytokines WJ-MSC evoked a significant decrease in the number of ED1 positive cells (from 100% after co-culture with WJ-MSC to 24.1% decrease after co-culture with WJ-MSC activated with IL-1β). Although the immunomodulative effect of WJ-MSC or stimulated with proinflammatory factors WJ-MSC was not significant. 

It is important to underline that the organotypic hippocampal slice culture reflects only local immune response and does not address the recruitment of peripheral inflammatory cells.

## 4. Discussion

Mesenchymal stem/stromal cells have unique immunomodulatory properties and can play an important role in the treatment of inflammatory and autoimmune diseases. Despite ongoing clinical trials and several years of follow-up after treatment, there are still concerns about their administration in the acute disease phase, which is due to insufficient knowledge on the behavior of cells in a stressful environment, including the active phase of inflammation.

As is often emphasized [30,31,32], the oxygen concentration is an important factor affecting cells at the time of inflammation. Eltzschig and collaborators underlined that hypoxia is a prominent feature of the inflammatory microenvironment and actively affect multiple cells, changing their migratory properties and directing from the oxygen-rich bloodstream to the hypoxic inflammatory milieu. Some groups even described that in in-vitro studies, hypoxic conditions could promote a proinflammatory M1 macrophage polarization [33,34,35,36]. Hence, we performed the experiments comparing the effects of 21% vs. 5% oxygen concentration on basic parameters of WJ-MSCs.

The phenotype of MSCs is identified by the absence of the CD34 and CD45 hematopoietic cell markers and the expression of CD73, CD90 and CD105. Majumdar et al. showed that a 2% oxygen concentration induced loss of typical mesenchymal markers (CD73+, CD90+ and CD105+) in the MSCs, although it was also associated with a long-term culture in vitro. The loss of the above-mentioned markers correlated with the cell senescence [37] and a secretory activity loss. In our experiments, under both 21% and 5% oxygen conditions, WJ-MSC treated with strong proinflammatory agents (IFNγ + TNFα and IL-1β) retained their mesenchymal immunophenotypes: constant expression of CD73+, CD90+ and CD105+ markers, without any shift towards hematopoietic markers. Moreover, our previous experiments [38] and other research groups’ reports indicated that cell cultivation in 5% oxygen was the factor that slowed down cell senescence, increased clonogenicity and maintained a young cell phenotype [39]. Concerns of cell culture in low oxygen were associated with possible cell rejuvenation and a risk of tumorogenicity [40].

Stimulation with inflammatory factors also did not induce changes in WJ-MSC morphology, with the cells retaining the fibroblast-like shape. In 2017, Klinker’s team developed a test correlating the MSCs immunosuppressive capacity with their shape. Accordingly, MSCs that retained an elongated, spindle shape were more predisposed to inhibit the inflammatory response than small cells with a round shape [41]. Other groups described that both in atmospheric conditions and <2% of O_2_ concentration, spindle-shaped morphology after treatment with the mixture of cytokines TNFα, IFNγ and IL-1β positively correlated with the amount of secreted anti-inflammatory factors, i.e., IL-1RA, IL-4 and IL-10 [42].

The rate of cell proliferation, proper cell cycle and maintenance of the secretory properties are important elements that correlate with MSC therapeutic properties. Stem/progenitory cells in healthy tissue, inside the niche, remain in a quiescent state, whereas in stress conditions and tissue injury they transit from deep quiescence to a quiescent G0/G1 state, with increased proliferative and regenerative potential [43]. 

In our experiments, both the exposure of MSCs to pro-inflammatory factors and 5% oxygen caused the cells to shift towards S and G2/M phases. Based on their in-vitro observations Khamchun and Thongboonkerd [44] concluded that cells involved in the repairing mechanisms underwent cell cycle shift from G0/G1 to S and G2/M phases. 

In our experiments, 5% O_2_ concentration did not cause significant changes in particular phases of the cell cycle, although the cycle time curtailed compared to standard conditions. As a result, the population doubling time (PDT) was shortened. The present results confirmed our earlier findings [38] and these of Obradovic et al. [41], who additionally noted that cell culture in 21% of oxygen resulted in increased number of apoptotic and necrotic cells as compared to the presence of 3% O_2_ [45]. Lavrentiev et al. described increased proliferation capacity of MSCs derived from an umbilical cord even in 1.5% oxygen concentration [46]. Numerous authors have confirmed the amplifying effect of decreased oxygen concentration on the proliferative activity of MSCs, and thus a higher number of population doublings.

Some researchers indicate a higher potential for differentiation of cells in reduced oxygen concentration, the finding which was not confirmed in our studies [47,48]. 

Our results confirm that the reduced oxygen concentration is critical and to some extent beneficial not only for the cell fate, but also for its motility. Moderate hypoxia or physioxia (5% oxygen) at a site of inflammation considerably raise the cells’ migration capacity.

Even a short-term (6- to 15-h) reduction in oxygen concentrations improve cells migratory potential in vitro [49]. However, the chemokines and chemotactic cytokines are also critical for the migration and positioning of cells. TNFα, IFNγ and cytokines belonging to the IL-1 group create the inflammatory microenvironment by activating and mobilizing the immune system [50,51]. The authors explained how IL-1/TNFa-NF-kB signaling axis functions as a component of an inflammation-associated niche.

As a result of the inflammatory response, the immune cells migrate from the periphery to the site of an injury. The reaction relies on chemotactic agents released locally, cytokine concentration and hypoxia [52]. In recent years, many authors reported a similar (both in-vivo and in-vitro) capacity of MSCs to migrate towards the site of damage [53,54]. Our research confirms the above reports and shows that WJ-MSCs are able to migrate towards the inflammatory microenvironment. Exposure to IFNγ, TNFα and IL-1β results in increased taxis in the direction of the agent. The effect is further enhanced by aerobic conditions. Lejis et al. reported that MSCs under hypoxia showed no changes in the adhesion and migration receptors at the molecular level, while the cells exposure to inflammatory factors resulted in increased MSCs migration [55].

Some authors postulate that to obtain MSCs migration, the cells have to be primed. IL-1β emerged as the most potent factor in stimulating cell migration. A similar observation was made by the Carrero group, which showed that IL-1β activates a set of genes associated with cell survival, adhesion, chemokine production and migration in an in-vitro model [52]. Moreover, these studies showed that IL-1β did not increase cell proliferation but accelerated their migration. These observations confirmed our results, in which IL-1β did not significantly affect PDT, but activated migration rate and extended its distance.

Recent clinical studies have shown that MSCs administered in the allogenic system act as a source of stimulating and trophic factors coordinating tissue repair through adjuvant properties [56,57]. Activation of immunomodulatory properties among MSCs requires their stimulation through the signals inputting from the local microenvironment. This way, activated cells are able to secrete paracrine factors changing the response of immune cells. IFNγ, TNFα and IL-1β are all shown to be involved in the regulation of this response [21,58,59]. Moreover, IFNγ, TNFα, interleukin-17 (IL-17), and IL-1β lead to the class I/II MHC and costimulatory molecules upregulation, cell proliferation, increased survival and enhanced immunomodulatory and immunosuppressive functions [60]. Similarly to IL-1 β, IFNγ is the one that can prime MSCs in vitro. While IL-1β, may prime MSCs alone as it does not require combinations of cytokines, to obtain the same effect IFNγ needs the presence of other proinflammatory cytokines, such as IL-1β or TNFα [22].

Our analysis of the culture medium following cell stimulation with proinflammatory cytokines shows the secretion of anti-inflammatory factors, such as IL-4 and IL-10. In the literature, these factors were associated with the mobilization of macrophages in the organism, and their polarization to the anti-inflammatory profile—M2 [3]. Another factor secreted by activated WJ-MSCs is the cytokine CCL2, responsible for attracting the eosinophil fraction to the site of inflammation.

IL-6, whose secretion increases after stimulation, has both anti- and pro-inflammatory properties, depending on the phase and nature of the disease. MSCs co-cultured with macrophage caused their polarization in the direction of the anti-inflammatory phenotype through IL-6 secretion [61]. Islam and colleagues described that mesenchymal stem cells derived from afterbirth tissues, exhibited strong immunomodulatory properties, secreted larger amounts of IL-6 than cells isolated from adult tissues (fat and bone marrow) [3]. IL-1β was the only tested factor to evoke CXCL10 secretion in MSCs. Considering that CXCL10 is the antagonist of IL-1Ra [42], this mechanism could explain a suppression of IL-1β being mediated by MSCs. Stimulation with all the examined factors resulted in an increase of IL-12 concentration, causing NK and macrophage activation, mainly also IFNγ secretion [62]. 

Overall, WJ-MSCs exposed to strong stress agents (5% oxygen and cytokines) neither lose their mesenchymal properties nor show proliferative disturbances, but they retain their directed migration and adjuvant properties. At the same time, we emphasize the crucial role of aerobic conditions on MSCs proliferation, rejuvenation and changes in the cell cycle. Although the impact we describe was beneficial, breaking a certain boundary may lead to the expression of pluripotent genes and compromise safety of the therapy. While observing the dual immunomodulatory effect of MSCs (secretion of both pro- and anti-inflammatory cytokines), we must remember that MSCs therapy may lead to both transient inflammation or suppression. Despite unequivocal results in preclinical studies, a transient fever (a marker of transient inflammation) after MSCs administration was reported in the clinical studies [63]. Also, the viral reactivation (herpesvirus), after MSCs administration was reported [18].

MSCs were also reported to promote human hepatocellular carcinoma metastasis under the influence of inflammation [64]. The described effect was imitated in vitro with the supernatant of MSCs pre-treated with IFNγ and TNFα. Treatment of cancer cells with the supernatant lead to epithelial-mesenchymal transition (EMT) and the effect was correlated with TGFβ secretion by cytokines stimulated MSCs.

To summarize, due to incoherent results, together with an insufficient number of preclinical studies focused on the influence of acute aggressive inflammation to MSCs’ fate, an explicit conclusion cannot be made at the moment.

## Figures and Tables

**Figure 1 jcm-10-01813-f001:**
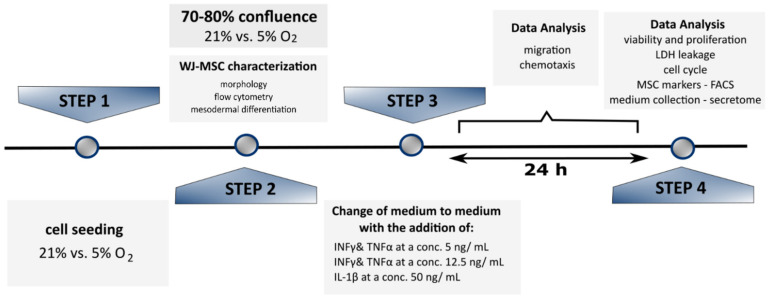
A schema of the experiments performed with selected cytokine variants.

**Figure 2 jcm-10-01813-f002:**
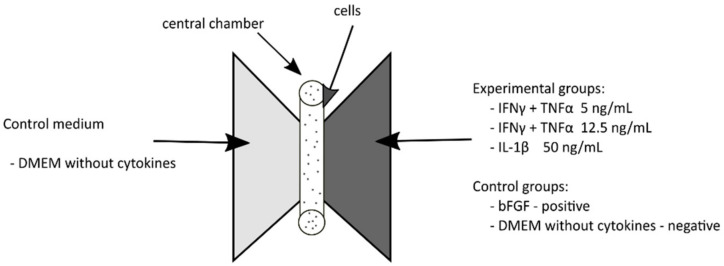
A schematic chemotaxis assay. Left chamber contained basic culture medium (DMEM), right chamber contained medium with tested factors (bFGF—positive control, DMEM without cytokines-negative control or proinflammatory factors) and central chamber with WJ-MSC.

**Figure 3 jcm-10-01813-f003:**
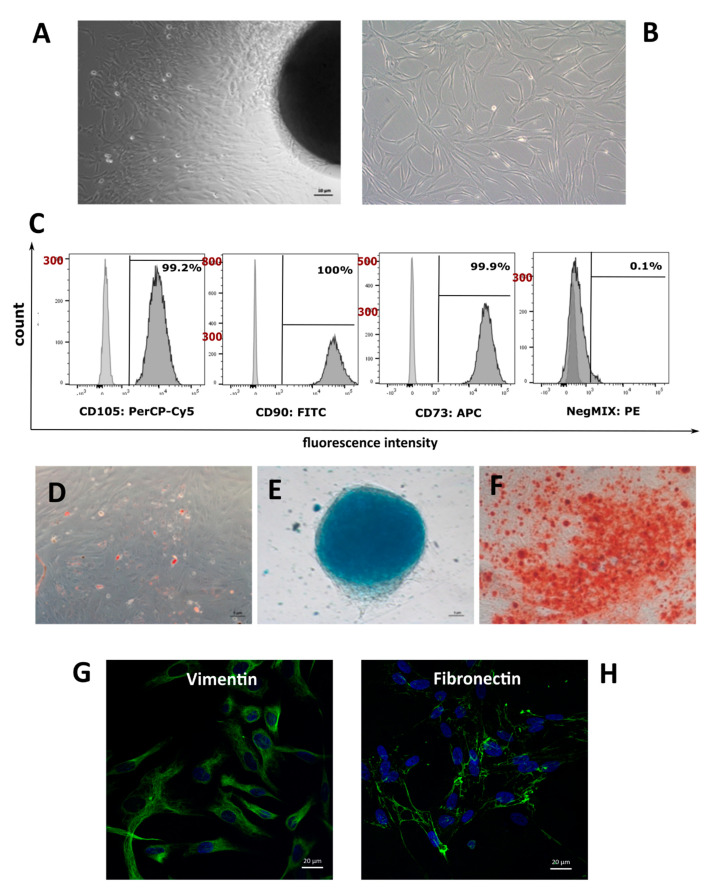
Characterization of non-treated WJ-MSCs. (**A**)—WJ-MSCs migration out of the explant derived from Wharton’s jelly. (**B**)—a fibroblast-like, adherent WJ-MSCs culture, with elongated, cylindrical shape morphology. (**C**)—Flow cytometry analysis. The number of cells expressing CD determines the peak in the lower right quadrant. The latter analysis showed culture with a relatively clear expression of specific mesenchymal markers (CD73, CD90, and CD105)—more than 98% positive markers. Not more than 1% of the WJ-MSCs expressed negative markers (CD34, CD11b, CD19, CD45, and HLA-DR). (**D**)—adipogenesis—positive Oil Red staining (red fat drops), (**E**)—chondrogenesis: positive Alcian blue staining for the presence of cartilage glycosaminoglycans, (**F**)—osteogenesis: positive Alizarin Red staining (**G**,**H**): immunocytochemical analysis of WJ-MSCs. Cells exhibited of typical cytoskeleton markers expression Vimentin (green-**G**) and Fibronectin (green-**H**).

**Figure 4 jcm-10-01813-f004:**
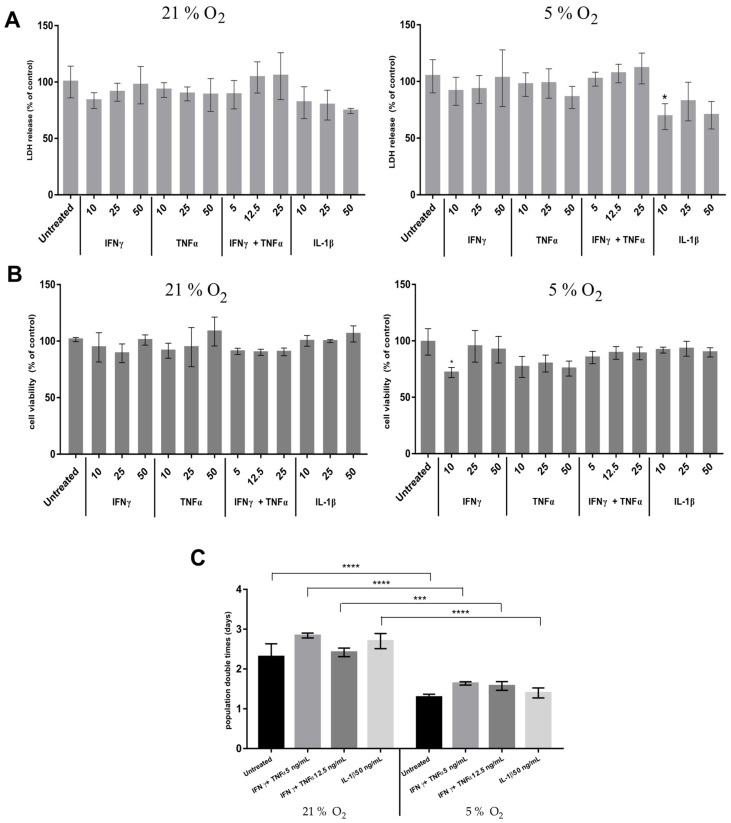
(**A**)—Results for WJ-MSC cell-membrane integrity analysis after treatment with proinflammatory cytokines in 21% or 5% O_2_ concentrations. There is no increase in LDH secretion into the medium by IFNγ and TNFα-treated cells, as well as a slight reduction in LDH secretion into the medium following IL-1β treatment in both aerobic variants. * *p* < 0.1 (**B**)—WJ-MSC viability analysis after treatment with proinflammatory cytokines. No significant differences in cell viability. (**C**)—Comparison of the WJ-MSC (21% O_2_) and WJ-MSC (5% O_2_) population doubling times. The results are mean values (one-way ANOVA) from 3 experiments ± SD; *** *p* < 0.001, **** *p* < 0.0001.

**Figure 5 jcm-10-01813-f005:**
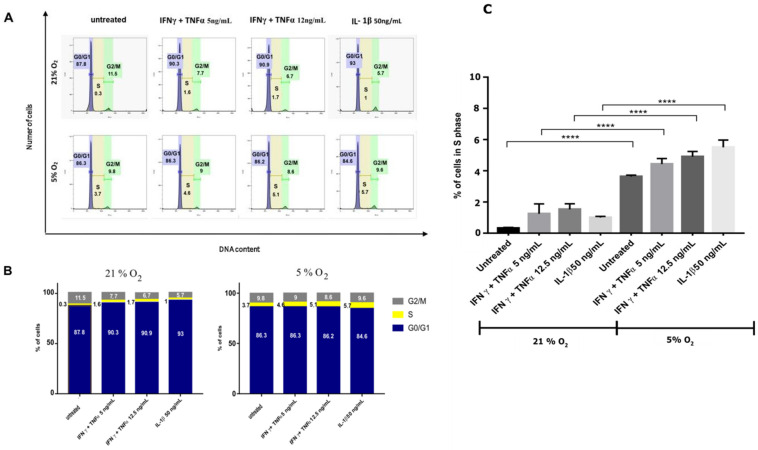
Cell-cycle analysis. (**A**)—Histograms were used to analyze the cell cycle in WJ-MSCs after treatment with proinflammatory cytokines under 21% or 5% O_2_ concentrations. (**B**)—Chart shows the percentage of cells in individual phases of the cell cycle at all tested groups. The results are mean values from 3 independent experiments in min. 3 replicates. (**C**)—chart shows the percentage of cells in S -phase of the cell cycle at all tested groups. The results are mean values (one-way ANOVA) from 3 independent experiments in min. 3 replicates ± SD **** *p* < 0.0001.

**Figure 6 jcm-10-01813-f006:**
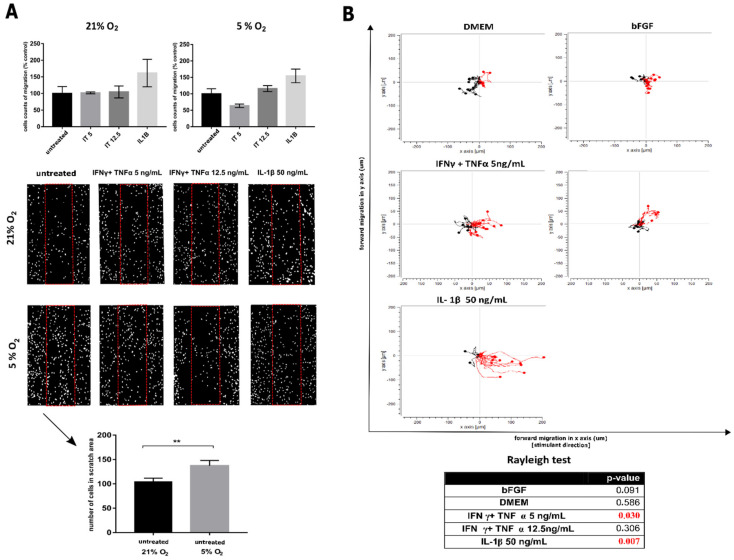
Migratory activity of WJ-MSCs. (**A**)—scratch assay analysis. IL-1β-treated cells at a concentration of 50 ng/mL displayed the greatest capacity for migration in culture under both oxygen concentrations. Under the conditions of 5% O_2_, control cells migrated faster than that cultured in 21% O_2_. The results are mean values (one-way ANOVA) from 3 independent experiments in min. 3 replicates ± SD, ** *p* < 0.01. (**B**)—chemotaxis. WJ-MSCs did not exhibit motility in the control environment. However, slightly increased migration towards the bFGF attractant was observed. Presence of the proinflammatory cytokines in the environment was the strongest factor enhncing migration (red line—movement towards cytokines). The strongest effect was observed when IL-1β was added to the medium. Table with results Rayleigh test. Statistically significant results are marked in red. In IFNγ and TNFα in concentration 5 ng/mL and IL-1β-treated cells distribution inhomogeneous. The results are min 20 cells from 3 different umbilical cords.

**Figure 7 jcm-10-01813-f007:**
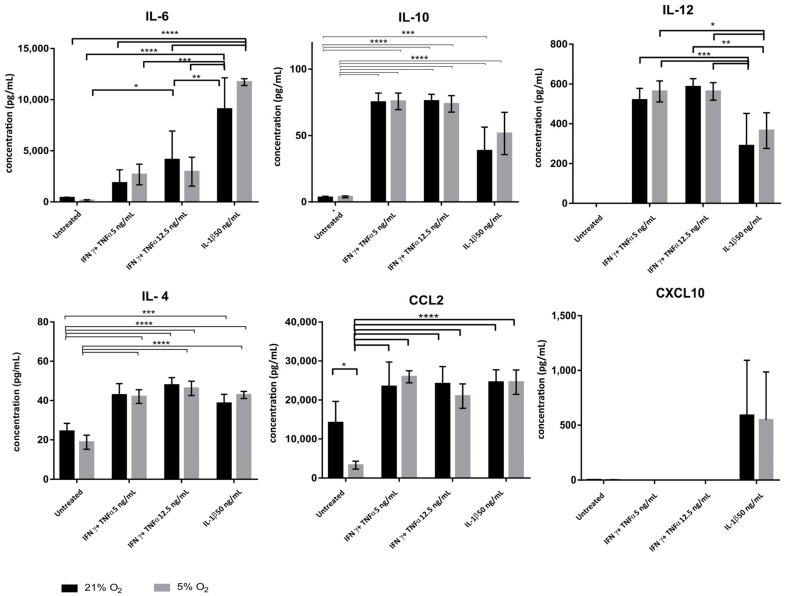
Quantitative analysis of cytokines secreted by WJ-MSCs cultured in 21% vs. 5% oxygen concentration after treatment with proinflammatory cytokines. Results are mean values (two-way ANOVA) for 3 independent experiments in min. 3 replicates ± SD; * *p* < 0.1, ** *p* < 0.01 *** *p* < 0.001, **** *p* < 0.0001.

**Figure 8 jcm-10-01813-f008:**
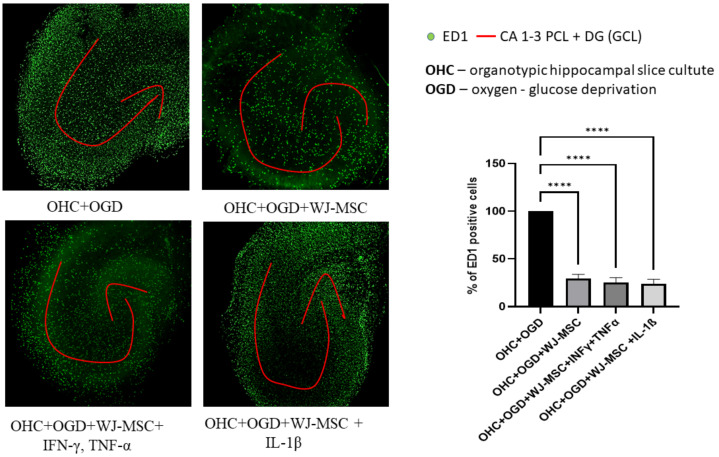
Immunomodulatory effect of WJ-MSC ex vivo. Increase in number of microglia in rat hippocampal organotypic slice culture (OHC) 24 h after OGD as indicated by cell labelling with ED1 marker (green). Representative images of injured rat hippocampal organotypic slice culture (OHC) used for microglia quantification. Co-culture with WJ-MSC evoke decrease of ED1+ cells. The effect is independent from WJ-MSC pretreatment with proinflammatory cytokines. Results are mean values (one-way ANOVA followed by Tukey’s test) for 3 independent experiments ± SD; **** *p* < 0.0001.

## Data Availability

The data presented in this study are available on request from the corresponding author.

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
