# Peer review of "The Effect of Proinflammatory Cytokines on the Proliferation, Migration and Secretory Activity of Mesenchymal Stem/Stromal Cells (WJ-MSCs) under 5% O2 and 21% O2 Culture Conditions"

_jcm, 2021, doi:10.3390/jcm10091813_

Round 1
Reviewer 1 Report
I thank the authors for their resubmission.
I appreciate that the authors have attempted to address several of my comments, albeit incompletely. Unfortunately, there are errors that do persist in the manuscript.
These include (but are not restricted to):
Figure 5: the values displayed in the cell cycle analysis plots for for Go/G1, S and G2/M are different to the values in the bar plots beneath.
Figure 6: While the authors state no significant differences were observed in their cover letter, a table is provided showing statistical differences.
Minor comment: there are several spelling mistakes throughout the manuscript body and legends.
Reviewer 2 Report
The authors studied Wharton Jelly MSCs’ proliferation, migration and secretion in two different oxygen concentration. They concluded that in a pro-inflammatory environment the WJ-MSC keep their typical surface molecule expression profile and proliferation rate but increase their migratory activity. They also increase the secretion of anti-inflammatory cytokines (IL4 and IL10). Based on their above findings the authors feel it is safe to use WJ-MSCs in therapy.
There is a lot of work and a lot of information in this paper and interesting and important findings that will potentially be useful for the field. However, there are some major and a few minor problems that need to be addressed/corrected before the work could be published.
The introduction could also be significantly shortened – it almost feels like a full review.
Major problems:
- Line 269 “The secreted amounts of IFNg and TNFa were estimated after subtraction of the amounts added to the medium for cell stimulation“
The authors say that pro-inflammatory cytokine primed MSC will secrete the same cytokines (TNFa and INFg) – something that nobody else has reported. There may be a serious technical problem here. After the stimulation the authors measure the concentration and subtract the added amount – suggesting that the difference must have been produced by the MSCs. Since this is a unique observation, the authors need to be very careful. They should do the experiment by incubating the MSCs, then removing the medium that contains the cytokines and wash the cells thoroughly. Then let the cells secrete whatever they secrete and measure the medium. In addition, they should measure mRNA in the cells and show that after the stimulation they can actually detect the message encoding TNFa and INFg and to show that it changes according to the dose they used.
- Line 272 hippocampal slice culture – first of all, the slice culture is very different from a real life ischemia, especially when immune function is studied. When there is hypoxia, inflammatory cells invade (not just microglia) and this changes the whole cytokine environment in the area of the brain studied (here it is hippocampus). This needs to be in the Discussion.
In the Result section (between lines 486-503; Fig.7) the images show a different story than the graph next to them. The primed WJ-MSCs seem to decrease the green staining (ED1) much more than the unprimed MSCs. The graph does not show this difference. Also, (at line 498 Fig.7) the image shows that blue is Ki67. The legend says blue is Hoechts. There is no mention of Ki67 in the whole paper. Was it used to show proliferating microglia? Was it not used at all? What is the blue in the images?
Minor problems:
- Throughout the paper in the Introduction and Discussion the authors need to clarify which species the papers that they refer to talks about. Which are human data and which are mice data? What kind of MSC was used? What is in vivo and what is in vitro? They should also try to refer to original articles about macrophage and MSC polarity changes.
- Reference 13 is Alzheimer’s disease, not lupus
- Reference 17 (line 70) uses adipose derived MSCs
- Reference 23 (line 105) the paper on the HPA axis is about viral infection (this should be pointed out)
- In the Methods, the primary antibodies used need to be described (species? Catalog number?); the type of secondary antibodies also need to be described (Fab fragment? Whole IgG?). What kind of controls were used for the ICC?
Round 2
Reviewer 2 Report
I am satisfied with the changes/corrections the Authors made. The only remaining small problem is that they need to spell/language check the Supplementary material that has several mistakes in it.
This manuscript is a resubmission of an earlier submission. The following is a list of the peer review reports and author responses from that submission.
Round 1
Reviewer 1 Report
Priming approaches to improve efficacy of MSC therapies has been reviewed (https://doi.org/10.1186/s13287-019-1224-y) with the effect of IFN-g and IL-1b on WJ-MSC previously characterised (Noone et al., Stem Cells Dev 2013, Carrero et al., Stem Cells Rev 2012). Here Figiel-Dabrowska et al., present a study assessing the phenotype of wharton jelly derived mesenchymal stem cell in response to IFN-g, TNF-a and IL-1b in normoxic and hypoxic conditions.
The choice of cytokines to assess, especially in combination IFN-g and TNF-a, is appropirate given prior data regarding Sars-Cov-2 associated illness (Karki et al., Cell Sept 2020). Assays describing generation, immunophenotypic characterisation and differentiation of WJ-MSC were performed appropriately. In general, the authors' conclusions regarding WJ-MSCs appear reasonable that in the setting of hypoxia and select inflammatory cytokines, although several major and minor comments need to be addressed. This is particularly important regarding description and presentation of data, including for biological replicates, review of figure legends, and re-evaluation of several points made in the discussion to ensure they are supported by the data presented.
Major comments:
1. Intrinsic variability has been noted in WJ-MSC (Paladino et al., Stem Cell International, 2017) How many indepedent WJ derived MSC were tested in each assay for Fig. 3, Fig. 4, Fig. 6. Fig. 7.?
2. In addition: Please state the number of biologic and technical replicates and provide a measure of uncertainty for Fig. 5., Fig. 6., Fig. 7.
3. Assays testing IFN-g and TNF-a independently should be presented with each experiment throughout this manuscript.
4. Scratch assay Fig. 6 : please define K. Please provide statistical analysis/comment: especially re: IL-1b treated cells - is the mobility statistically significant?
5. Chemotaxis Assay Fig. 6B : Please define the axes. Please provide statistical analysis comparing migration of bFGF compared to DMEM (ie. did the positive control work?), IFN-g & TNF-a at 5ng/mL, 12.5 ng/mL and IL-1b 50ng/mL. How were the concentrations of IFN-g & TNF-a in this assay? Why does the IFN-g & TNF-a at 5ng/mL, 12.5 ng/mL directionality in the axes appear different? Please provide migration data for single cytokine with dose titration for IFN-g & TNF-a.
6. Discussion: the statement "exposure of MSCs to pro-inflammatory factors and 5% oxygen caused the cells to shift towards S and G2/M phases (line 481): this currently isn't supported by the presented data (Fig 5) which has no replicates presented or statistical test applied.
7. Discussion: the statement "WJ-MSCs are able to migrate towards the inflammatory microenvironment. Exposure to IFN-g, TNF-a and IL-1b results in increased taxis in the direction of the agent" requires more statistical justification to support this comment (Fig. 6b) especially for IFN-g/TNF-a exposure (see comment 5).
8. Discussion: "Our analysis of the culture medium following cell stimulation with proinflammatory cytokines shows secretion of anti-inflammation factors .... is responsible for a mobilisatoin of macrophages in the organism (line 537-539)". Please present this data.
Minor comments.
1. Method 2.5 please define "GS"
2. Method 2.8 & 2.10: why were different cytokine (rather than single cytokine) combinations used for IFN-g and TNF-a compared with IL-1b? How were concentrations chosen?
3. Formatting error 518-519
Reviewer 2 Report
In this study, Wedzinska et al study the effects of hypoxia and inflammatory cytokines on MSC properties. The authors did a nice job of validating MSC phenotypes (Figure 3). However, this study lacks novelty. It could also benefit from some of the additional experiments mentioned below.
Major Points
- In the abstract, the authors claim: “It is concluded that the acute inflammation does not cause phenotype changes or give rise to genetic instability of WJ-MSCs, and nor does it inhibit the secretory properties providing for their use against acute infections.” The data do not support this conclusion. The authors used some conditions (e.g. hypoxia, cytokines) to simulate “acute inflammation,” though the lack of in vivo work does not allow them to make this conclusion. Also, they did not measure “genetic instability,” but rather cell cycle distribution. Genetic instability could be measured by any of the following: mitotic errors in fixed immunofluorescence or live cell imaging, degree of aneuploidy (i.e. from chromosome spreads).
- In the abstract, the authors discuss MSC polarization. I would like to know how hypoxia and inflammatory cytokines affect MSC polarization.
- The authors should assess how hypoxia and cytokines affects MSC immunosuppression, for example with a T cell suppression assay.
- Page 7, lines 283, the authors mention that these cells appear more senescent. They should assess beta-galactosidase activity to confirm.
- The scratch assay is a good first test to assess migration, though it does not control for cell proliferation. The authors could redo this experiment in the presence of a cell cycle inhibitor (e.g. aphidicolin, thymidine) to eliminate the effects of proliferation on apparent migration. The better experiment would be to track single cell migration using live cell imaging.
Minor Points
- Page 2, line 57-58, this is just one sentence and should not be its own paragraph.
- Page 2, line 59, grammar is off, I would change it to “The use of MSCs in treating autoimmune diseases such as lupus erythematosus and rheumatoid arthritis has resulted in…”
- Page 2, line 63, change “efficient” to “effective.”
- Page 2, line 91, I’m not sure what you mean by the “natural immunological border.” Recommend rephrasing.
- Page 2, line 95, remove colon and use commas.
- Figure 3A, what is the dark round object on the right of the image? I think a different image should be chosen without this.
- Page 9, lines 325-334, these paragraphs should be combined into one.
- What is the conclusion from section 3.2? Recommend stating this on page 10 line 350.
- Page 10, line 360, change “stimulants” to “cytokines.”
- Page 11, line 363, “cytokines did not act to inhibit the life functions of cells.”
- Figure 5B, there should be statistics to show the differences you mention in the text.
- In Figure 6, why did you use multiple concentrations of TNF and IFNg but only one concentration of IL1b?
- Throughout the figures, I often see the letter “K.” Does this mean control? I would spell out “control” or “untreated.”
- What is the conclusion from section 3.5? Recommend stating this on page 14 line 441.
- Combine paragraphs on lines 452-466.
- Line 462, typo with “reaserch”
- Line 498, should be “extent” rather than “extend”
- Lines 519-520, there is a gap for some reason, needs to be removed.
- Lines 573-575, I am not sure what you are trying to say here. What do you mean by “incoherent results”? The use of “aggressive” to describe inflammation is also confusing.
Round 2
Reviewer 1 Report
Response to Major Revision
The authors have made several changes to the manuscript which have clarified aspects of the figures and discussion.
However, several errors in the revision remain. Unfortunately, this includes an explanation in the author response (Major Comment 5) regarding chemotaxis assay directionality which would not generally be acceptable in a final publication, and conflicts with the data axes presented.
There are also statements in the author response which aren't reflected in the data presented (Major Comment 6) where no measure of uncertainty is presented.
The authors undertaking three biological and three technical replicates should provide data points to identify the sample origin and present standard error of the mean between each origin source. The data points for which the two way ANOVA applied, and the post-test comparison used to derived P values described.
Reviewer 2 Report
I appreciate the efforts made by the authors to revise the manuscript. In particular, I’m glad to see the inclusion of single cell migration data. However, I still think additional experiments are needed to make this work suitable for publication.
Major Points
- If the authors are going to make conclusions about “genetic instability,” they should show additional data to suggest that, as I had mentioned previously, such as with the following experiments: mitotic errors in fixed immunofluorescence or live cell imaging, degree of aneuploidy (i.e. from chromosome spreads).
- Please provide additional citations and language to verify that hypoxia and the cytokines used mimic the "inflammatory milieu," as the authors stated in their response.
- Thank you for providing the in vivo data from brain. Including these data would strengthen this paper. Would be best to add quantification as well.
- Please include these beta gal data in the paper with quantification.
- Please add a new conclusion sentence to the abstract to replace the one that was deleted.
- Please add a hypoxia condition to the migration data (Fig 6B).
- The results of the new migration data (Fig 6B) are not discussed in the text.
